# Current Knowledge on Exosome Biogenesis, Cargo-Sorting Mechanism and Therapeutic Implications

**DOI:** 10.3390/membranes12050498

**Published:** 2022-05-06

**Authors:** Shenmin Xie, Qin Zhang, Li Jiang

**Affiliations:** 1National Engineering Laboratory for Animal Breeding, Key Laboratory of Animal Genetics, Breeding & Reproduction, Ministry of Agriculture, College of Animal Science & Technology, China Agricultural University, Beijing 100193, China; s20203040583@cau.edu.cn (S.X.); qzhang@cau.edu.cn (Q.Z.); 2College of Animal Science and Technology, Shandong Agricultural University, Tai’an 271018, China

**Keywords:** exosomes, biogenesis, cargo sorting, ESCRT, microRNA, bioengineering

## Abstract

Extracellular vesicles (EVs) are nanoscale membrane vesicles released by donor cells that can be taken up by recipient cells. The study of EVs has the potential to identify unknown cellular and molecular mechanisms in intercellular communication and disease. Exosomes, with an average diameter of ≈100 nanometers, are a subset of EVs. Different molecular families have been shown to be involved in the formation of exosomes and subsequent secretion of exosomes, which largely leads to the complexity of the form, structure and function of exosomes. In addition, because of their low immunogenicity and ability to transfer a variety of bioactive components to recipient cells, exosomes are regarded as effective drug delivery systems. This review summarizes the known mechanisms of exosomes biogenesis, cargo loading, exosomes release and bioengineering, which is of great importance for further exploration into the clinical applications of EVs.

## 1. Introduction

The earliest research on extracellular vesicles was performed in the 1960s, when scientists coincidentally discovered that different tissues [1], cells [2] and body fluids [3,4] can secrete certain vesicles into the external environment.

The extracellular vesicles initially observed were directly budding from the plasma membrane. In the 1980s, Pan BT and Harding C [5,6] discovered a more complex secretion pattern of extracellular vesicles from mature reticulocytes. Studies have shown that some extracellular vesicles in reticulocytes are formed by inward budding of the inside of the cell endosome accompanied by the formation of multivesicular bodies (MVBs) that envelop these newly formed pockets. MVBs can fuse with the plasma membrane and release internal small vesicles into the external environment. Due to the particularity of this extracellular vesicle, Johnstone, a pioneer in this field, named it an “exosome” for the first time in 1987 [7]. Since this discovery, extracellular vesicles derived from endosomes have been found to be released from antigen-presenting cells [8,9], epithelial cells [10] and tumor cells [11].

Discoveries lead to scientific questions. The first question asked about exosomes is: how are exosomes formed? Currently, the classical process of exosome biogenesis involves three main steps: the establishment of the endosomal system, the formation of intraluminal vesicles (ILVs), and the fusion of multivesicular bodies (MVBs) with the plasma membrane. This overall “construction roadmap” has basically become consensus in this field, but the “minutiae” are still not well understood. For example, changes in the physical structure of the membrane upon budding and the identification of substances with sorting functions that are critical for the recruitment of cargos remain unknown. In addition, intracellular MVBs may not be exactly the same, and there may be multiple heterogeneous subgroups of MVBs. This viewpoint has been proven by an increasing number of studies. For example, the cholesterol content in MVBs is not exactly the same among B lymphocytes, and only MVBs with a high cholesterol level eventually form exosomes [12]. In addition, there is evidence that the exosomes secreted from the apical and basolateral sides of polarized cells have a different composition [13,14]. This heterogeneity of exosomes can be reflected in their size, material composition and function [15].

With the discovery of mRNAs and microRNAs in exosomes and their transfer to cells [16], increasing attention has been directed to the important roles of exosomes. Accumulating evidence has demonstrated that exosomes play important roles in cell-to-cell communication [17,18], organism development [19], immune response [20], neurons [21] and tissue repair [22]. In addition, many studies have found that the secretion rate of exosomes in certain pathological cells (including tumor cells) is often higher than that of normal cells, indicating that exosomes may participate in certain disease processes and play roles in promoting tumor progression [23] and assisting the spread of certain viruses [24].

With the development of mass spectrometry technology and various kinds of omics, many studies on the transcriptome, proteome, lipidome, and metabolome performed to characterize exosomes have aimed to analyze the composition of exosomes accurately [25,26] (Figure 1). These studies provide a solid research foundation for the study of the biogenesis, release and cargo sorting of exosomes. Notably, the molecular substances involved in these processes may not exist in exosomes, as they likely detach from exosomes after performing their functions [27]. This review summarizes the recent advances made in understanding the molecular mechanism of exosomes formation, cargo sorting, exosomes bioengineering and the clinical application of exosomes.

## 2. Characterization of Exosomes

According to the size of extracellular vesicles (EVs) and their biogenesis, EVs can be roughly categorized into three types [18,28,29]: (1) Apoptotic bodies, about 1–5 μm in diameter, are formed by apoptotic cells in the form of plasma membrane budding. (2) Microvesicles, about 100–1000 nm in diameter, are formed by direct outward budding from the plasma membrane. (3) Exosomes, about 30–150 nm in diameter, are produced by the fusion of multivesicular bodies produced through the endosomal pathway with the plasma membrane.

Although we have classified extracellular vesicles, it is difficult to separate and characterize the subpopulations of EVs in practical applications. Even exosomes isolated from a single cell line present with differences in morphology and size. Moreover, the density of exosomes is affected by the ratio of protein to lipid; therefore, the abundance and type of cargo affect the density of exosomes may exceed the typical 1.1–1.2 g/mL range [8]. Since it is difficult to determine the subcellular origin, particle size and specific markers of extracellular vesicle subgroups, the exosomes isolated and characterized by existing methods may include different subtypes or have other unique features that we typically ignore. Therefore, it is a challenge to establish a scientific and accurate method for exosome purification and analysis.

However, the accuracy of exosomal characterization continues to improve with the advancement of various technologies. Since the first ultracentrifugation method was used to isolate exosomes from the reticulocyte culture medium in 1987 [30], new separation and purification methods have been continuously explored and improved to overcome specific problems. For example, exosomes obtained by ultracentrifugation are easily contaminated by coprecipitated exosomes components [31], and to solve this problem, many improvements have been made to this method, such as the development of sucrose [32] or iodixanol [33] density gradient centrifugation methods, which are based on particle density. Some size-based separation techniques, such as ultrafiltration and size-exclusion chromatography (SEC), have also been established. Ultrafiltration relies on ultrafine nanomembranes that are categorized according to different molecular weight cutoff values, which are used to separate exosomes. Currently, two mature ultrafiltration devices are used to separate exosomes: tandem-configured microfilters and sequential ultrafiltration systems can effectively separate particles in the range of 20–200 nanometers. Size-exclusion chromatography can be used to separate exosomes according to their permeability in a stationary phase. The advantage of this technique is that the integrity, structure and biological activity of the vesicles are retained [34]. In addition to these two conventional methods, some new size-based separation techniques have been recently developed. A group studying the separation of exosomes in clinical samples (including plasma, urine and lavage fluid) obtained from cancer patients developed a size-based exosomes isolation tool called exosome total isolation chip (ExoTIC), which is a modular platform that helps obtain a high yield of highly pure exosomes from biological fluids [35]. In another recent study, a protocol for the use of asymmetric-flow field-flow fractionation (AF4) technology was developed and optimized to rapidly separate exosomes with intact biophysical properties from extracellular nanoparticles (ENPs) [36]. AF4 technology separates particles of different sizes and molecular weights through the interaction of two particle flow streams (horizontal and longitudinal). Due to the influence of cross flow, the elution speed of large particles is slower, while that of small particles is faster [37].

In addition to the aforementioned methods based on physical properties, immunoisolation methods have been developed [38] that can effectively improve the capture efficiency and specificity of exosomes through the specific binding of antigens and antibodies. Recently, an exosome-specific dual-patterned immunofiltration (ExoDIF) device was developed through the combination of ultrafiltration and immunoaffinity technology [39]. The device is composed of two different immune mode layers. By generating a mechanical vortex, the chance of antibodies and exosomes binding is increased, which leads to high-throughput exosomes separation and high-resolution specificity. Technical devices designed to separate exosomes on the basis of their combined physical and biochemical characteristics have been continually developed, greatly improving the purity and yield of the product.

In addition, some innovations in the identification of exosomes have also made it possible to understand exosomes accurately. For example, the emergence of atomic force microscopy and cryo-electron microscopy has partially solved the problem of exosomal size measurement errors caused by sample processing through techniques such as nanotracking particle analysis (NTA) and resistive pulse sensing (RPS). Super-resolution microscopy can be used to determine the size of exosomes and the distribution of cargos [40]. Single-particle interference reflection imaging combined with conventional fluorescence microscopy can be used to detect the presence and abundance of specific lipids, proteins, nucleic acids and carbohydrates in exosomes [41]. Moreover, a high-sensitivity flow cytometry (HSFCM) instrument was developed for the quantitative multiparameter analysis of single exosomes, which provides a sensitive platform for the detection of surface proteins in individual exosomes [42]. In the future, more technologies will be applied to this field, which will not only improve the accuracy of the purification and characterization of exosomes but also lead to the development of new standards for the classification of exosomes.

## 3. Exosomes Biogenesis and Cargo Sorting

In recent years, the mechanisms driving the formation and secretion of exosomes have started to be revealed. In the classical pathway, exosomes are formed in endosomes. First, the plasma membrane invades the cytoplasm and then forms a cup-shaped structure that is either independent of or fused to a pre-existing early endosome, forming a new early-sorting endosome (ESE). Then, part of the ESE membrane invades the inner endosome and buds to generate multiple intraluminal vesicles (ILVs), forming MVBs. MVBs further develop into late-sorting endosomes (LSEs). Nature MVBs formed at this time fuse with the plasma membrane and finally release the internal ILVs into the external environment.

In addition to the construction of the physical structure of the exosomal membrane, the above-mentioned processes also include the sorting, ingestion and transportation of cargos. Exosomes with normal physiological functions can only be produced when these two processes are synchronized and stabilized. The studies described below are summarized in Table 1 and Table 2.

**Table 1 membranes-12-00498-t001:** Research on the biogenesis and release of exosomes.

Protein	Material Used in Study	Used for Exosome Definition	References
**ESCRT-Dependent**			
Hrs	DCs, HeLa-CIITA	MHC-II, VPS4B, Tsg101, CD63, HSC70, CD81	[43,44,45]
STAM1	HeLa-CIITA	CD63, CD81, MHC-II, HSC70	[43]
Tsg101 (VPS23)	HeLa-CIITA, MCF-7, DCs, MDCK	CD63, CD81, MHC-II, HSC70, syndecan-1, ALIX	[43,45,46,47]
CHMP4C (SNF7C)	HeLa-CIITA	CD63, CD81, MHC-II, HSC70	[43]
CHMP4B (SNF7B)	HeLa-CIITA	TSG101, RAB5, HRS	[48]
Alix	HeLa-CIITA, MCF-7, DCs	CD63, CD81, MHC-II, HSC70, syndecan-1, TSG101, RAB5, HRS	[43,46,48]
VPS4	HeLa-CIITA, MCF-7, DCs	CD63, CD81, MHC-II, HSC70, syndecan-1	[43,46]
Syntenin	MCF-7	CD63, HSP70	[46]
Syndecan	MCF-7	CD63, HSP70, Alix	[46]
**ESCRT-Independent**			
nSMase2	Oli-neu, HEp-2	PLP, Hrs, Tsg101	[49,50]
PLD2	RBL-2H3, MCF-7	Syntenin, ALIX, CD63, SDC1CTF	[46,51,52]
DGKα	J-HM1–2.2	CD63, β-Actin, Fasl	[53]
CD9	HEK293, BMDCs	β-Catenin, Flotillin-1	[54]
CD82	HEK293	β-Catenin	[54]
CD63	HEK293, Rat1, HK1, DG-75, MNT-1, HeLa	HSC70, Calnexin, CD81	[55,56]
RAB31	HEK-293T, HeLa	Flotillin-1, Flotillin-2, CD9, CD81, CD63, Tsg101, Alix	[57]
**Exosome Release**			
RAB11	K562, Drosophila S2	Transferrin receptor, Lyn, HSC70, Evi	[58,59,60]
RAB27a/b	HeLa-CIITA, Human peripheral blood, 4T1	CD63,Tsg101, Hsc70, Hsp70, VLA-4, Hsp90, Alix	[61]
RAB35	HepG2	CD63, Tsg101	[62]
RalA, RalB	4T1	ALIX, CD63, HSC70, TSG101	[63]
VAMP7	K562	Acetylcholinesterase activity	[64]
YKT6	A549	Tsg101	[65]
Tetherin	HeLa	CD63, ALIX, TSG101	[66]

ESCRT: endosomal sorting complex required for transport.

### 3.1. The Formation and Maturation of the Endosomal System

#### 3.1.1. Formation of Early-Sorting Endosome

The biogenesis of exosomes begins with the endosomal system, and the endosomal system begins with early-sorting endosomes (ESE). In mammals, the ESE is the first programmed site of membrane-related cargo sorting and solute transfer. An ESE can consist of a variety of receptors (cell signaling receptors, nutrient transporters, ion channels, adhesion molecules, and polarity markers), lipid membranes and extracellular fluid.

An ESE is an relatively mature early “large vesicle” formed by the fusion of primary endocytic vesicles [83]; therefore, in essence, an ESE is a primary endocytic vesicle. Currently, there are two main explanations for the formation of endocytic vesicles: clathrin-mediated endocytosis (CME) and clathrin-independent endocytosis (CIE). The CIE pathway mainly includes caveolar, CLIC/GEEC, and ARF6-dependent pathways [84]. These endocytic pathways may undergo different forms of envelopment with or without specific protein coating, but they all perform the “mission” of the phagocytosis of molecules. The CME pathway was the first to be discovered and the best-characterized endocytic pathway, which includes steps such as nucleation, cargo selection, clathrin coat assembly, vesicle scission and uncoating [85]. The process of generating endocytic vesicles through the CME pathway involves the initial recruitment of relevant protein to form a clathrin-coated pit (CCV), and then, these coated vesicles separate from the mother membrane. Finally, the HSC70 protein and auxiliary protein auxilin (or cyclin G-related kinase (GAK) in non-neural tissues) remove the mesh coating to reveal primary endocytic vesicles, which then move to the target endosome and fuse with it.

The ESE formed upon vesicle fusion includes a tubular structure and vacuole structure, which both differ substantially in material composition, evolutionary direction, and physiological functions. The tubular extension contains molecules such as RAB5 [86], RAB4, RAB7 [87], RAB11, Arf1/COPI, Retromer and Caveolae-1 [88], which are involved in sorting cargos for delivery to different targets. The cargos recruited to tubular vesicles by these proteins mainly include dissociated ligands and solutes and proteins internalized from body fluid components. These substances can be returned to the plasma membrane in various ways and can be reused. The current research suggests that the recycling routes back to the plasma membrane can be categorized into two types [89]: “fast recycling” and “slow recycling”. “Fast recycling” refers to several small vesicles containing material to be recovered directly separating from the ESE tubular structure and being directly transported to the plasma membrane [90]. “Slow recycling” refers to ESE tubular vesicles budding and splitting to form a cystic/tubular structure, which is transported to recycling endosomes (Res), and then, the released vesicles are transported along the microtubules to the plasma membrane [91]. The vacuole structure of an ESE further forms late-sorting endosomes (LSEs) in a subsequent process.

#### 3.1.2. The Formation and Maturation of Late-Sorting Endosome

As mentioned above, an ESE contains a tubular structure and a vacuole structure, but only the vacuole structure can form a late-sorting endosome (LSE) in the subsequent process, which means that the tubular structure needs to be discarded during the formation of an LSE. The mechanism of the process is not fully understood. Current research suggests that before the formation of an LSE, RAB5 in the vacuole area of an ESE recruits RAB7 to form a transient heterozygous endosome, and then, RAB5 is converted into the GDP-bound form and dissociates together with its effector. At this time, a vacuole structure containing only RAB7 is produced. This embryonic vacuole leverages various sorting mechanisms to continuously recruit cargo, and at the same time, the membrane surface begins to bud inward to form multiple small intraluminal vesicles, which gradually separate from the tubular structure of the early endosome to form separate transport carriers called multivesicular body (MVB)/endosomal carrier vesicles (ECV) [87,92]. In this process, dynamin provides a source of power for the separation of the tubular structures and vacuole structures [93]. The MVBs/ECVs, which “drop” from the ESE, continue to undergo changes and further mature into complete LSEs. This process involves several important events, including RAB protein conversion [92], acidification [94], microtubule-dependent movement [95], morphological changes and phosphatidylinositol (PI) conversion [96].

### 3.2. Formation of MVBs/ILVs and Cargo Sorting

After the formation of an ESE, the endosomal membrane begins to bud inward to form multiple small intraluminal vesicles and ultimately form MVBs. The formation of MVBs/ILVs is essential for the selective sorting of membrane-related cargo, and ILVs are direct intracellular manifestations of exosomes. Generally, MVB/ILV formation pathways are divided into two types: the ESCRT (endosomal-sorting complex required for transport) complex-dependent pathway and the ESCRT complex-independent pathway.

#### 3.2.1. The ESCRT Complex-Dependent Pathway

The ESCRT complex is the most important member of the MVB/ILV generation machinery. The ESCRT complex is composed of five different protein complexes, including ESCRT-0, -I, -II, -III and accessory proteins (notably, the AAA ATPase Vps4 complex) [97]. The respective members of each complex have been identified through a series of genetic screenings with yeast [98,99,100]. ESCRT-0, -I, and -II, as early ESCRT elements, all contain ubiquitin-binding domains (UBDs), which are involved in the selective sorting of ubiquitinated molecules. Although the late ESCRT components ESCRT-III and AAA ATPase Vps4 have no ubiquitin-binding domain and cannot be used as scaffolds for cargo recruitment, they play important roles in inward budding, vesicle shedding and the cargo isolation of ILVs after vesicle formation. The influence of the ECSRT complex on the biosynthesis of MVBs/ILVs is primarily explored by interfering with ESCRT complex protein activity and observing the changes in the secretion of ILVs by cells. Recently, in a relatively comprehensive RNAi screening study [43], seven ESCRT proteins (CHMP4C, VPS4B, VTA1, ALIX, Hrs, TSG101 and STAM1) that affect exosome secretion were identified. Further research revealed that the deletion of the Hrs, TSG101 and STAM1 proteins in ESCRT-1 resulted in a decrease in exosome secretion. In addition, other similar studies have shown that the deletion of ESCRT-0 proteins Hrs [44] and TSG101 [45] is directly related to the synthesis of ILVs and the secretion of exosomes.

MVBs/ILVs also transport various cargos during the formation process. The ESCRT complex plays a decisive role in this process. According to the characteristics of the cargos, the mechanisms by which ESCRT complexes identify and sort cargos can be generally divided into ubiquitination-dependent pathways and ubiquitination-independent pathways.

Early ESCRT components (ESCRT-0, -I, and -II) play important roles in the identification and sorting of ubiquitinated cargoes. The Hrs subunit of ESCRT-0 contains the ubiquitin-interacting motif (UIM), and the STAM1 subunit contains two ubiquitin domains in UIM and VHS, both of which can recruit ubiquitinated cargo to the endosomal membrane. The two core subunits of the ESCRT-I complex, Tsg101 [101] and UBAP1 [102], each contain a ubiquitin-binding domain, which acts as a receptor for ubiquitinated cargo to be identified and sorted. The ESCRT-II complex is a heterotetrameric protein complex composed of Vps36, Vps22 and Vps25 [103], but only the GLUE (Eap45 in mammals) domain of Vps36 can interact with phosphatidylinositol 3-phosphate 3 (PI3P) to identify ubiquitinated cargo [104]. Notably, there is an extraordinary level of coordination between ESCRT UBDs, and they collectively recognize a diverse set of cargos and do not act sequentially with discrete steps [105]. Through its own FYVE domain, the Hrs subunit in ESCRT-0 binds to PI3P on the restriction membrane to target MVBs, and it recruits ESCRT-I by interacting with Tsg101, which recruits Gal3 to ILVs for apical exosome-mediated release upon recognizing and binding a highly conserved tetrapeptide P(S/T)AP motif [47]. Finally, the Vps28 subunit of ESCRT-I interacts with Vps36 of ESCRT-II to form a large ESCRT component for the systematic and orderly sorting of cargos.

The ubiquitinated cargos identified and captured by the early ESCRT complex are initially spread on the surface of the restricted membrane. To prevent these cargos from spreading laterally throughout the membrane, ESCRT-III components, including SNF7, are assembled into spiral oligomers around the cargos. This oligomer can stabilize cargo in the designated area and provide the inward tension required for membrane invagination [106]. The recruitment of ESCRT-III in yeast follows two parallel pathways: one depends on ESCRT-I/II (the canonical pathway), and the other depends on ESCRT-0/BRO1 (the ALIX homolog in yeast) [107]. In mammals, CHMP4B (one of the ESCRT-III components) recruitment to endosomes is coordinated by the dual actions of the ALIX BRO1 domain interacting with CHMP4 and LBPA [48]. In addition, the direct interactions between ALIX, syntenin and syndecan-4 are essential for proper enrichment of the ESCRT-III machinery at the abscission site, which can help maintain the ESCRT-III polymer at the abscission site until the final cleavage [108]. After abscission, the ESCRT complex is dissociated under the action of AAA ATPase Vps4, forming the free ESCRT elements, which are ready for the next round of sorting work.

In recent years, a large number of studies have shown that the products sorted by the ESCRT complex are not limited to ubiquitinated molecules. Some membrane proteins that do not undergo ubiquitination also enter ILVs. Interleukin-2 receptor β (IL-2Rβ) directly binds to the C-terminus of Hrs in ILVs in a ubiquitination-independent manner. Similarly, through its V domain, the ESCRT-related protein ALIX can bind to the YPX(3)L motif of protease-activated receptor-1 (PAR1) [109] and purinergic receptor (P2Y1) [110]. In addition, a recent study showed that the C-terminal portion of pX (hepatitis A virus structural protein) can interact with ALIX similar to the interaction between the YPXnL “late domain” and the V domain of ALIX, thus loading large protein complexes into exosome-like EVs [111]. In addition, many studies have shown that syndecan (SDC) heparan sulfate proteoglycans interact with syntenin-1 and ALIX to form the syndecan/syntenin/ALIX pathway and control exosome biogenesis. Syntenin is a small intracellular scaffold protein. It contains ZO-1 (PDZ) domains that bind to the C-terminus of SDC or SDC cargo (such as FGF:FGFR complexes) and an N-terminal domain that binds to ALIX via three LYPxL motifs [46]. The PDZ domains of syntenin can also bind to several other membrane proteins, including CD63. Interestingly, disruption of the syndecan–syntenin–ALIX axis does not affect the size or number of exosomes but reduces the amount of exosomal syntenin, ALIX and SDC; other exosomal markers are unaffected [112]. Heparanase, a modulator of the syndecan–syntenin–ALIX pathway, promotes endosomal membrane budding and the biogenesis of exosomes by trimming the heparan sulfate chains on syndecans [113].

In summary, the ESCRT complex plays an important role in the formation of ILVs. With this huge “working machine”, ILVs can effectively detach from the restrictive membrane of the MVB and enter the cavity while recruiting related cargo to enter, forming a complete intracavitary vesicle structure.

#### 3.2.2. ESCRT Complex Independent Pathway

With the continuous deepening of research, it has been found that the formation of MVBs/ILVs does not entirely depend on the ESCRT complex. An increasing number of studies have shown that ILVs can be formed in cells with all ESCRT-related subunits removed [114], which indicates a mechanism independent of the ESCRT complex that also controls the formation of MVBs/ILVs or exosomes.

Because the physiological characteristics of lipids are critical to the remodeling of cell membranes, many scientists first turned their attention to the role of lipids in the formation and secretion of exosomes. Stuffers et al. found that the release of exosomes derived from mouse oligodendrocytes is related to the activity of neutral sphingomyelinase (nSMase, a ceramide-producing enzyme) [49]. Synthetic ceramide can directly promote the budding of MVBs and can also be metabolized to produce sphingosine 1-phosphate (S1P), which binds to the S1P receptor on MVBs to promote the production of ILVs [115]. In another study, it was found that the overexpression of phospholipase D2 (PLD2) was enriched in exosomes derived from RBL-2H3 cells [51]. Furthermore, PLD2 was found to be the effector protein of small GTPase ADP ribosylation factor 6 (ARF6), which controls the budding of MVBs and the development of exosomes [52]. In addition, the study by Mazzeo et al. showed that diacylglycerol kinase α (DGKα) mediates the maturation and secretion of MVBs by regulating the subcellular localization and activation of PKD1/2 [53].

Another ESCRT-independent pathway for exosome production and cargo sorting is closely related to tetraspanins. Tetraspanins, constituting a protein superfamily, organize membrane microdomains by forming clusters and interacting with a large variety of transmembrane and cytosolic signaling proteins; these domains are known as tetraspanin-enriched microdomains (TEMs) and serve as cargo transportation platforms [116]; For example, the melanocyte-specific glycoprotein PMEL enters ILVs upon interacting with CD63 [55]; membrane metalloproteinase CD10 interacts with CD9 to enter ILVs; CD9 and CD82 interact with E-cadherin to promote the exosome secretion of β-catenin [54]. CD63 also plays an important role in the transport of latent membrane protein 1 (LMP1) in exosomes [56]. Moreover, these supramolecular complexes in cell membranes not only control the transportation of exosomal cargo but also affect the production of exosomes. A previous study showed that exosome release from dendritic cells generated in CD9-knockout mice is diminished [54]. Interestingly, another study showed that tetraspanins CD81 and CD82 regulate the formation and/or development of cell membrane protrusions, which undoubtedly provided new ideas for studying the mechanism of exosome budding [117].

In addition, a recent study revealed a novel ESCRT-independent mechanism in cells [57]. The research team found that EGFR, which was expected to sorted into lysosomes for degradation, was present in exosomes derived from the serum of cancer patients. This discovery suggests that the pathway for EGFR sorting into exosomes may not rely on the ESCRT complex. The group further found that nonubiquitinated EGFR can be endocytosed by cells under serum starvation conditions and colocalized on CD63-positive MVBs formed under the action of RAB31. Specifically, RAB31 binds to the SPFH domain in effector FLOTs after activation, induces membrane budding of MVBs through the flotillin domain, and finally forms ILVs. Interestingly, the study also found that EGFR can phosphorylate of RAB31 at tyrosine at three positions, Y76, Y86 and Y137, in effect activating RAB31. Similarly, other receptor tyrosine kinases (RTKs) can activate RAB31 through phosphorylation to promote its entry into exosomes. There was no effect on this process after removing the components of ESCRT, such as Hrs, Tsg101 and Alix.

In summary, there are two pathways for the formation of MVBs/ILVs or exosomes (including biogenesis and cargo loading): the ESCRT-dependent pathway and the ESCRT-independent pathway. The ESCRT-dependent pathway can be subdivided into ubiquitin-dependent pathways and ubiquitin-independent pathways. Together, these complex mechanisms provide a reliable mechanism for the formation of exosomes to ensure that this important physiological process can proceed normally under different conditions. Notably, these mechanisms are not completely independent. In contrast, different exosome formation pathways may be related to each other (Figure 2). Ceramide and the ESCRT complex exert a coordinated effect and promote the inclusion of phospholipase 3 in exosomes [118]. In the syndecan–syntenin–ALIX pathway, knocking out nSMase partially impaired its ability to promote exosomal production [46]. However, a recent study reported that the syndecan1–syntenin1–ALIX machinery and the sphingomyelinase-dependent ceramide production machinery independently mediate apical exosome release and basolateral exosome release, respectively [119]. The specific mechanisms involved in this process are still not fully understood.

Notably, the formation of ILVs depends not only on the interaction between molecules but also on the abundance of incorporated cargo. Theoretically, all the biological mechanisms of transport vesicles would have negative feedback regulatory mechanisms to prevent the formation and accumulation of empty vesicles.

### 3.3. Release of Exosomes

After their formation, MVBs face two “fates”: (1) transport to lysosomes and the degradation of MVB contents after membrane fusion or (2) upon fusion with the plasma membrane, ILVs are released to form exosomes. As mentioned earlier, there are different subgroups of MVBs in cells, each of which determines its own “fate”, but the molecular mechanism of this regulatory event is unclear. A study has shown that ISGylation, a post-translational ubiquitin-like modification, promotes the fusion of MVBs with lysosomes, thus leading MVBs to the degradation pathway and away from the secretory pathway [120]. In recent years, considerable evidence has demonstrated that tetraspanins play important roles in exosome production. However, a recent study found that the concentration of TSPN6 determines the localization of MVBs: low levels of TSPN6 enable the secretion of exosomal syntenin and SDC4, and high levels of TSPN6 cause the same cargos to undergo lysosomal degradation [121]. These results indicate that there might be some mechanism that promotes the export of goods to exosomes by inhibiting the fusion of MVBs and lysosomes in the body.

Furthermore, some studies have found that the RAB protein and its effector protein/adapter protein play significant roles in the transport route of MVBs. Studies have shown that the RAB7 adaptor protein RILP (RAB-interacting lysosomal protein), which is critical for linking RAB7-positive vesicles to dynein motor complexes, is cleaved after inflammation [82] or viral infection [122], to form a RILP cleavage fragment (cRILP). This cleavage directly abrogates the binding of GTPase RAB7 to the dynein motor complex and promotes kinesin-mediated movement to the cell surface and finally completely changes the transport direction of vesicles [82]. However, the exact mechanism by which cRILP mediates the transformation of RAB7 from binding to the dynein motor complex to binding to kinesin is still unknown. The following explanations have been proposed: (1) the formation of cRILP causes RAB7 to bind to another effector protein, such as FYCO1 [123], which regulates the peripheral transport of vesicles through the RAB7 interaction with the kinesin motor complex or (2) RAB7-positive vesicles may be connected to motor proteins through another GTPase adaptor system that binds to kinesin after RILP is cleaved and then relocates RAB7-containing vesicles to the periphery of the cell. In addition, Kang et al. found that exogenously expressed RAB31 inhibits RAB7 activity (it is believed that RAB7 mainly mediates the fusion of MVBs and lysosomes) by recruiting the GTPase activating protein TBC1D2B to the surface of MVBs/LSEs, which promotes the fusion of MVBs containing EGFR with the plasma membrane [57].

The transport of MVBs to the plasma membrane is a prerequisite for the release of exosomes, which requires the participation of actin and the microtubule cytoskeleton [124]. Microtubule skeletons are similar to intracellular “tracks”, along which MVBs are transported to the periphery of the cell at the plasma membrane. Actin is the source of power to propel MVBs to their destination accurately. It was found that invadopodia formation, the structure of which is formed by an aggressive form of actin, has a synergistic effect with exosome secretion [125]. Another study showed that cortactin, a kind of cortical actin, controls the transport and plasma membrane docking of MVBs in cancer cells [126].

MVBs transported to the periphery of the plasma membrane need to fuse with the plasma membrane to release ILVs from the lumen to the extracellular space. This function is mainly performed by the SNARE proteins, RABs and Ras GTPase proteins [127].

#### 3.3.1. The SNARE Proteins

The SNARE proteins consists of one R- and three Q-SNAREs, and each component forms a “helical coil” [128]. The process of SNARE-mediated fusion of vesicles with the target membrane exhibits “zipper-like action” [129,130]. When the membranes fuse, one R-SNARE constitutes v-SNARE, and three Q-SNAREs constitute t-SNARE. The two elements are anchored on the donor and acceptor membranes to form anti-SNARE complexes (SNAREpins). This process produces an inward force F, which increases incrementally from the N end of the membrane to the C end of the membrane, pulling the two-layer membrane together and forcing them to fuse. Many studies have shown that the SNARE protein is involved in the release of exosomes. Overexpression of the N-terminal domain of R-SNARE vesicle-associated membrane protein 7 (VAMP7) in human leukemia cells inhibits the formation of SNARE complexes and thus reduces the release of exosomes [64]. When the expression of another R-SNARE protein, YKT6, is decreased, the number of exosomes secreted by human lung cancer cells is also decreased [65]. In Drosophila S2 cells, the reduction in Q-SNARE synaptic fusion protein 1A (Syx1A) resulted in the obstruction of exosome secretion [131].

#### 3.3.2. RABs and Other Ras GTPases

The first RAB protein discovered to be involved in the release of exosomes was RAB11. The overexpression of RAB11 mutant protein in K562 cells resulted in decreased exosome secretion [58]. In follow-up studies, calcium ions were also found to be involved in this process [59], and a ca~(2+)-dependent SNAP receptor and RAB-binding protein have been identified in cancer cells; both promote the fusion of MVBs and plasma membrane in a RAB11-dependent manner [60]. Upon identifying 59 RAB GTPases using shRNA in HeLa cells, five RAB proteins were found to be involved in exosome secretion, among which RAB27a and RAB27b play a role in the docking fusion between MVBs and the plasma membrane [61]. The study showed that after RAB27a was silenced, the size of MVBs increased significantly, and after RAB27b silencing, MVBs redistributed to the perinuclear area. This conclusion has also been verified in different cell lines [125,132,133]. In a study of Caenorhabditis elegans, it was found that Ras-related GTPase homolog Ral-1 and SNARE protein SYX-5 colocalized on the plasma membrane. The decreased expression of the latter causes MVBs to accumulate in the plasma membrane and cannot dock with the plasma membrane. At the same time, researchers also found that the homologs of Ral-1 in mammals, RalA and RalB, are also necessary for cultured cells to secrete exosomes [63].

In addition, recent studies have found that certain lncRNAs can indirectly affect membrane docking and fusion by affecting the expression and localization of the RAB protein. The lncRNA HOTAIR affects the colocalization of VAMP3 and SNAP23 by regulating the expression and localization of RAB35 and ultimately promotes exosome secretion [62]. It was also found that lncRNA PVT1 promotes the docking of MVBs with the plasma membrane by changing the expression and localization of RAB7 and regulating the colocalization of YKT6 (a v-SNARE homolog) and VAMP3 [134].

According to the existing theoretical model, after the fusion of MVBs with the plasma membrane, exosomes are released, leaving the maternal cell to perform their functions at new destinations. However, a recent study found a class of exosomes on the cell surface of HeLa cells, showing remarkable similarity with HIV and other viruses [66]. This study used gene-editing technology to remove tetherin (which can bind the HIV virus to the cell surface) in HeLa cells and found that the originally clustered exosomes quickly separated and vacated the cell surface. This outcome suggests that there may be a regulatory mechanism controlling whether exosomes are released for short-range communication or long-range communication.

### 3.4. miRNAs Sorting into Exosomes

Exosomes include proteins, lipids, DNA, mRNAs, and noncoding RNAs. As important noncoding RNAs, miRNAs have attracted considerable attention. MiRNAs are endogenous noncoding RNAs with a length of approximately 18–24 nucleotides, and they play important roles in gene expression regulation. Some studies have shown that miRNAs do not enter exosomes randomly but are incorporated through an internal selection mechanism to regulate cell functions. There are significant differences in the types and quantities of exosomal miRNAs derived from different cell sources, and the expression of exosomal miRNAs is also significantly different then the miRNA expression of maternal cells. Studies have shown that the expression levels of let-7 family members in exosomes derived from different cancer cell lines differ [135]. In addition, Guduric-Fuchs et al. measured the expression level of miRNA in a variety of cell lines and exosomes derived from these cells and found that miR-451, miR-150, and miR-142–3p are preferentially located in exosomes compared to the cells. The relative content of these microRNAs in exosomes was hundreds of times higher than that in cells. However, the knockout of AGO2 resulted in a higher relative content of these miRNAs in cells than in exosomes [136]. Moreover, many studies have also been focused on the changes in miRNA expression levels of exosomes from the same source under different physiological conditions [137,138]. Therefore, the inclusion of miRNAs into exosomes is a cell-autonomous process of selection.

Neutral sphingomyelinase 2 (nSMase2) is the first molecule reported to be involved in miRNA sorting into exosomes. There is evidence showing that nSMase2 can regulate the content of miR-210 in exosomes derived from cancer cells [67] and mesenchymal stem cells [68]. nSMase2 can also regulate the entry of miR-10b into exosomes derived from metastatic breast cancer cells [69].

Substance P (SP) and its receptor, NK1R, have been implicated in the expression of many miRNAs in the cytosol of human colonocytes [139]. Recently, SP/NK-1R signaling has been shown to regulate the sorting of exosomal miRNAs. The results of this study showed that the level of exosomal miR-21 produced by human colonocytes in response to SP signaling was increased compared with that produced by unstimulated cells [71].

In addition, some sequence characteristics of miRNA itself can also determine the fate of its entry into exosomes. Koppers-Lalic et al. showed that the distribution of miRNA in human B cell-derived exosomes is related to the post-transcriptional modification of its 3′ end and that 3′ end adenylation facilitates miRNA retention in the cell, while 3′ end urine glycation may promote the inclusion of miRNA into exosomes [70]. This conclusion was also reported in another study. Wani et al. revealed that increased post-transcriptional 3′-end uridylation of miR-2909 is the driving force for the recruitment of this miRNA to exosomes secreted by prostate cancer cells (PC-3) [140]. Furthermore, researchers believe that adenosine kinase is closely related to the addition of nontemplate nucleotides to the 3′ end of miRNAs, which affects the distribution of miRNAs in cancer cells and their secreted exosomes. However, the role of adenosine kinase in the post-transcriptional modification of miRNAs and their subsequent sorting need to be further explored.

Among many influencing factors, RNA-binding proteins (RBPs) are among the most important. RBPs facilitate the targeting of miRNAs to exosomes by binding to specific motifs in miRNAs.

#### 3.4.1. Heterogeneous Nuclear Ribonucleoproteins (hnRNPs)

Heterogeneous nuclear ribonucleoprotein (hnRNP) is a protein superfamily consisting of more than 20 proteins, which are named alphabetically from A1 to U. Currently, three types of proteins are known to play important roles in sorting specific miRNAs for entry into exosomes: heterogeneous ribonucleoprotein A2B1 (hnRNPA2B1), heterogeneous ribonucleoprotein A1 (hnRNPA1), and synaptotagmin binding cytoplasmic RNA interacting protein (SYNCRIP, also known as hnRNP-Q).

The hnRNPA2B1 protein was first found to bind miR-198 in exosomes after SUMOylation [72], and the specific binding motif was determined to be GGAG. In a recent study focusing on this mechanism, selectively inhibited miRNA packaging in exosomes was found to be inhibited by the function of hnRNPA2B1 [141]. However, the study also found a negative relationship between hnRNPA2B1 protein and miRNA sorting in exosomes. It has been shown that the knockout of hnRNPA2B1 protein increases the expression level of miR-503 in endothelial cell-derived exosomes, but there is no known binding motif between hnRNPA2B1 and this miRNA [142]. This finding indicates that there may be a pathway that is not yet fully understood that negatively regulates the level of miRNA in exosomes.

The HnRNPA1 protein, a ubiquitously expressed RNA-binding protein recognizing the motif UAGGG(A/U), has been reported to mediate the sorting of miR-196a [73] in exosomes derived from cancer-associated fibroblasts (CAFs) by binding to a specific motif (UAGGUA) in the 5′ end of miR-196a. Another study found that hnRNPA1 can promote miR-522 entry into CAF exosomes after deubiquitination by USP7 [143]. In addition to CAFs, hnRNPA1 can also sort miR-320 for inclusion into exosomes by recognizing specific motifs (AGAGGG) in leukemia cells [144].

The SYNCRIP protein, also known as the hnRNP-Q protein, is an important member of the hnRNP protein family. Santangelo L et al. found that the SYNCRIP protein binds to miR-3470a and miR-194-2-3p in exosomes and identified a specific binding motif (GGCU) [74]. Further studies have demonstrated that the NURR domain (an RNA recognition domain) in the SYNCRIP protein directly binds to the GGCU motif [145]. Destroying this domain weakens the binding of SYNCRIP to miR-3470 and reduces the level of miR-3470 in exosomes.

#### 3.4.2. Argonaute 2 (AGO2)

The AGO2 protein is a component of the RNA-induced silencing complex (RISC). It binds to mature miRNA and facilitates miRNA inhibition of gene expression through its endonuclease activity [146]. Previous studies have found that AGO2 can stabilize miRNA and protect miRNA in vesicles from RNase degradation [147]. A subsequent study showed that both AGO2 levels and AGO2 phosphorylation affect the inclusion of specific miRNAs into exosomes. Notably, this correlation is regulated by the KRAS/MEK/ERK signaling pathway [75]. Changes in this signaling pathway can affect the ability of AGO2 to sort let-7a, miR-100 and miR-320a for entry into exosomes [75].

#### 3.4.3. Y-Box-Binding Protein-1 (YBX-1)

Y-box-binding protein-1 (YBX-1) is a multifunctional regulator of gene expression that has positive or negative effects on transcription and translation [148]. In recent years, studies have found that the YBX-1 protein can specifically sort miR-133 for entry into exosomes derived from human endothelial progenitor cells (EPCs) induced by hypoxia/reoxygenation [149]. In addition, Shurtleff et al. designed a cell-free reaction system by adding the target miRNA to a separation membrane containing cytoplasm to simulate the process of exosome uptake of cargo in vitro and discovered that YBX-1 can sort miR-223 for inclusion into exosomes [76,77]. In addition, they found that YBX-1 promotion of miR-233 entry into exosomes does not depend on a specific recognition motif but proceeds through the interaction of its internal cold shock domain with miR-223. To confirm the specificity of this sorting mechanism, the same cell-free simulation method was used to evaluate the effect of YBX-1 on the packaging of miR-190, and the results showed that the sorting of miR-190 was not affected by YBX-1.

#### 3.4.4. Serine- and Arginine-Rich Splicing Factor 1 (SRSF1)

SRSF1 was originally identified as a splicing factor in eukaryotic cells, but it was later discovered that SRSF1 shuttles between the nucleus and the cytoplasm to regulate RNA metabolism, miRNA processing and other cellular events independent of the mRNA splicing process [150]. A recent study confirmed that SRSF1 binds to miR-1246 (the most abundant miRNA in exosomes derived from pancreatic cancer cells) and analyzed the RNA sequences of the miRNAs highly enriched in cancer exosomes and regulated by SRSF1. As expected, a 6 bp motif was found to be shared among 36 of the 45 miRNAs enriched in exosomes, including miR-1246 [78].

#### 3.4.5. Major Vault Protein (MVP)

Major vault protein (MVP) is a ribonucleoprotein involved in the transport of RNA from the nucleus to the cytoplasm [151]. Recently, a study [79] showed that MVP can transport miR-193a from tumor cells to exosomes. Knocking out MVP resulted in the accumulation of miR-193a in cells, not in exosomes, but it did not affect the content of miR-126a or miR-148a, indicating that MVP selectively combined with miR-193a to form an MVP protein–miR-193a complex, which was ultimately packaged into exosomes. In addition, another study specifically identified RBPs in exosomes that interact with different RNA species. Compared with the effect on five other candidate RBPs, post-transcriptional silencing of MVP resulted in a significant reduction in total exosomal shuttle RNA (esRNA), indicating that MVP plays a very important role in the transport of RNA to exosomes [152].

In recent years, the functions of other RNA-binding proteins closely related to miRNA sorting have been reported. For example, MEX3C can indirectly sort the target miRNA into endosomes by interacting with adaptor-related protein complex-2 (AP-2), which is a cargo adaptor in clathrin-mediated endocytosis [80]. It has been reported that the La protein binds to specific motifs (such as UGGA motifs) and selectively packages miR-122 into exosomes [81]. Furthermore, a recent study showed that fragile X mental retardation 1 (FMR1) is recruited to MVBs by cRILP (the cleavage product of the RAB7-trafficking adaptor protein RILP] and sorted miR-155 for entry into exosomes by recognizing the AAUGC motif [82].

## 4. Bioengineering of Exosomes

The natural sources and biological characteristics of exosomes benefit the application of exosomes, making them safe and effective delivery systems. According to the mechanism of exosome biogenesis, secretion and absorption, exosomes were artificially designed to meet special purposes [153]. This process is called exosome bioengineering. The driver of exosome biological modification behavior lies in the “messenger” properties of exosomes, which act as bridges between parent cells and recipient cells and deliver selectively loaded cargo to target cells. Currently, exosome modification research mainly involves two methods [154]: active approaches and passive approaches. The former include methods of incorporating target substances in the process of exosomal biogenesis, such as through the genetic modification of cells; the latter include methods of loading or combining exogenous substances after exosome secretion, such as electroporation or chemical conjugation. The key points of exosome bioengineering are the accurate loading of target cargos into exosomes and improving the targeting of exosomes (Figure 3).

Currently, exosomal cargo involves mainly small RNAs and proteins. The initial goal of exosome bioengineering was to load small RNAs into exosomes because many studies had shown that small RNAs can be transported to target cells through exosomes. These small RNAs loaded into exosomes mainly include miRNAs and siRNAs, which can regulate gene expression. In exosome engineering, miRNAs are incorporated into exosomes through passive methods, typically transfection. The approaches for loading siRNA into exosomes include nongenetic methods such as electroporation and surface coating, as well as active approach packaging with a pre-microRNA backbone [155]. However, the use of surface coating or electroporation has profound disadvantages: (1) siRNA on the surface of exosomes has difficulty entering the recipient cell, and a considerable portion is retained on the endosome or at cell surface. (2) The use of electroporation may cause siRNA precipitation, making it difficult to incorporate the siRNA into exosomes [156]. (3) The use of biological modification methods is likely to affect the membrane structure of the exosomes and destroy membrane proteins, thus reducing the targeting efficiency of the exosomes. Therefore, the biological modification method with a pre-microRNA backbone packaging is an important method for exosomal siRNA loading in the future. The pre-miRNA-451 used to describe this method is a “noncanonical” pre-miRNA that can bypass the Dicer enzyme and directly bind to Ago2 [157]. Notably, if the pre-miR-451 stem-loop structure is maintained, it can be reprogrammed with other miRNA sequences. Using this feature, siRNA can be incorporated into the variable 3′ end of pre-miR-451 packaged with the backbone of pre-miRNA-451 and delivered into exosomes together. In addition to small RNAs, proteins can also be loaded into exosomes through the application of exosome bioengineering. Previous studies reported that the inclusion of the tumor suppressor PTEN in exosomes requires the Ndfip1 protein containing a late domain, which is an adaptor protein of the Nedd4 family of E3 ubiquitin ligases [158]. Based on this idea, a recent study added a WW tag to the target protein Cre recombinase to promote its direct interaction with Ndfip1 and ultimate binding to the ESCRT complex through the ubiquitination of Nedd4-1 and entering exosomes [159].

Another key aim of exosomal bioengineering is the targeting of exosomes. Currently, this kind of research is based on the principle of specific binding of receptors and ligands to regulate the targeting of exosomes. The key is to enable exosomes to carry the ligands that can bind to receptors on target cells. Early studies were based on Lamp2b, a membrane protein on extracellular vesicles, fusing with rabies virus glycoprotein (RVG), which specifically binds to acetylcholine receptor 3 [160]. The chimeric protein was transfected into dendritic cells, expressed, and then incorporated into derived exosomes. Finally, exosomes modified with RVG were specifically delivered to neurons, microglia, and oligodendrocytes in the brain. On the basis of this study, another study used iRGD (pEGFP-C1-iRGD-Lamp2b) to replace the RVG fragment and obtain modified exosomes for the encapsulation of doxorubicin [161]. Other proteins involved in exosome biogenesis and ultimately remaining on the exosome membrane can also fuse with targeted ligands or receptors to promote tissue-specific targeting of exosomes. For example, in a recent study, a single polypeptide composed of two single-chain variable fragment antibodies targeting CD3 and EGFR was genetically linked to the TM domain of PDGFR [162]. Synthetic multivalent antibody-retargeted exosomes (SMART-Exos) can bind to T cells and triple-negative breast cancer (TNBC) cells.

However, the method of fusing targeting ligands with exosomal membrane proteins has drawbacks: On the one hand, it may cause damage to exosome membrane proteins; on the other hand, some fused targeting ligands will be degraded prematurely before they perform their targeting functions. For these reasons, Michelle et al. designed a fusion protein targeting peptide, Lamp2b, in which glycosylation motifs were added to various positions [163]. The introduction of this motif not only protected the peptide from degradation but also increased the overall expression of the Lamp2b fusion protein in cells and exosomes. In addition, Kooijmans et al. fused a nanobody encoding anti-epidermal growth factor receptor (anti-EGFR) as a targeting ligand for tumor cells with glycosylphosphatidylinositol (GPI) anchor signal peptides derived from decay-accelerating factor (DAF) [164]. The fusion protein did not change the general characteristics of EVs (i.e., morphology, size distribution or protein marker expression) and greatly improved the binding of EVs to tumor cells under static conditions.

## 5. Therapeutic Applications of Exosomes

As new therapeutic drug delivery systems, extracellular vesicles have received increasing attention from researchers. Many potential targets have been found to treat or delay disease progression, and some synthetic drug carriers have been extensively used, such as synthetic lipids, nanoparticles and other pharmaceutical drug vehicles. Compared with these synthetic drug carriers, exosomes have unique advantages. As mimics of “nature’s delivery systems”, exosomes can deliver drugs to target cells via membrane fusion or endocytosis, greatly reducing the incidence of immune rejection induced by foreign substances. In addition, compared to the use of maternal stem cells to treat related diseases, the use of exosomes can overcome a variety of biological barriers and prevent the development of tumors induced by stem cells.

Exosomes are heterogeneous. In addition to the existence of some conserved molecules critical for common biological activities, a large number of specific molecules in exosomes reflect the intrinsic characteristics of exosome-producing cells. Therefore, the selection of suitable exosome-producing cell lines is of great significance for basic research and clinical application.

### 5.1. Mesenchymal Stem Cell (MSC)-Derived Exosomes

Among the cell types known to produce exosomes, mesenchymal stem cells (MSCs) are ideal candidates for the mass production of exosomes for use in drug delivery.

As transport carriers, MSC-derived exosomes show an innate tendency to target inflammatory and tumor tissues preferentially in vivo [165]. Taking advantage of this feature, many studies have regarded MSC exosomes as potential therapeutic drugs. The role of exosomes in therapeutics is the distribution of their internal cargos, especially mRNAs and miRNAs, to target cells. For example, MSC exosomes can inhibit cytokine storms and reduce apoptosis and virus replication rates by delivering mRNA and miRNA to lung tissue [166]. Furthermore, some specific miRNAs in MSC exosomes have notably attenuated disease. A recent study showed that the anti-inflammatory microRNA miR-146a was strongly upregulated upon IL-1β stimulation and was selectively packaged into MSC exosomes [167]. Exosomal miR-146a was transferred to macrophages, resulting in M2 polarization, which eventually prolonged the survival time of septic mice. Another study showed that MSC-derived exosomes ameliorated atherosclerosis in ApoE−/− mice and promoted M2 macrophage polarization in plaques through the miR-let7/HMGA2/NF-κB pathway [168].

However, the MSC exosomes used to improve the pathological state of the body were not all “natural”. Most MSCs were artificially modified. Engineered MSC exosomes can be more easily manipulated to meet the designer’s research purpose. Some studies used transfection [169] or cell synthesis [170] to package into MSC exosomes substances that do not naturally exist in these exosomes or to add an endogenous MSC component at a level that exceeds the amount typically carried in natural MSC exosomes to endow these exosomes with enhanced functions. Another kind of research mainly used drugs to treat the human mesenchymal matrix/stem cell-like cell (MSC) population and then collected MSC-derived exosomes. The stimulus of the external environment may change the cargo-sorting mechanism of exosomes and lead to the appearance of heterogeneous exosomes. A recent study [171] showed that after treating several human mesenchymal stromal/stem cell-like cell (MSC) populations with sublethal concentrations of Taxol for 24 h, the MSC control exosomes showed almost no growth-inhibitory effects on tumor cells, but the exposed MSC-derived exosomes loaded with Taxol were associated with 80–90% cytotoxicity. In addition, the study revealed that the use of MSC-Taxol exosomes reduced the amount of distant organ metastasis observed in the lung, liver, spleen and kidney by at least 50%. The same method was used in the research of Ralf Hass [172], Virgínea [173], and others; these studies aimed to provide an unlimited source for large-scale MSC-derived exosome production with reproducible quality to enable differential drug targeting of tumors or other diseases. However, the exact mechanism of exogenously administered exosomes in vivo has not been fully elucidated and needs further exploration by researchers.

In addition, a considerable amount of preclinical data have proven the safety of exosomal treatments, and some clinical trials have been largely completed. A recent study on the treatment of COVID-19 explored the safety and effectiveness of allogeneic bone marrow mesenchymal stem cell-derived exosomes (ExoFlo™) in a treatment for severe COVID-19 [174]. In this study, 24 patients received a single 15 mL intravenous dose of ExoFlo, and the safety and efficacy were evaluated from days 1 to 14 after treatment. The results showed that ExoFlo has the ability to restore oxygenation, attenuate cytokine storms and rebuild immunity. It is a promising therapeutic candidate for the treatment of severe COVID-19. However, due to the lack of established cell culture conditions and production, a superior plan for the isolation and storage of exosomes, data on the best therapeutic doses and a dosing plan, and a reliable method of titer analysis for evaluating the efficacy of exosomes, the clinical application of MSC exosomes is limited.

### 5.2. Dendritic Cell (DC)-Derived Exosomes

DC-derived exosomes (Dexs) are used in tumor therapy because they carry all the molecules needed to activate the antitumor T cell-mediated immune response [175]. Similarly, relevant studies obtained antigen-specific DC-released exosomes through transfection, electroporation or viral transduction of DCs [176,177]. However, later studies found that immature dendritic cells have the inherent ability to take up proteins and peptides from the surrounding bodily fluids and tissues [178]. This finding indicates that proteins and peptides can be loaded into immature dendritic cell-derived exosomes without special loading techniques. Based on this characteristic, a study used model tumor antigens such as ovalbumin (OVA) to incubate immature dendritic cells to obtain OVA-loaded exosomes. However, the number of exosomes produced by immature dendritic cells is limited, which makes it difficult to produce the number of exosomes required to meet the needs for clinical applications [160].

In terms of clinical application, Dexs have been used for cancer vaccination and treatment [179]. To date, three phase I clinical trials and one phase II clinical trial have been completed [180]. As early as 2005, three phase I clinical trials were completed [181,182,183]. Exosomes were isolated from monocyte-derived dendritic cells and loaded with MAGE tumor antigens. The immune effect was evaluated after the patients were inoculated with the exosome vaccine. Finally, the safety, feasibility and effectiveness of the Dexs loaded with MAGE tumor antigen were proven. Then, one of the research groups carried out a phase II clinical trial to test the clinical benefit of IFN-γ-Dexs loaded with MHC class I- and class II-restricted cancer antigens that had been used in maintenance immunotherapy after induction chemotherapy in patients bearing inoperable non-small cell lung cancer (NSCLC) without tumor progression. This phase II clinical trial confirmed that Dexs enhanced the antitumor immunity function of NK cells in patients with advanced NSCLC [184].

### 5.3. HEK293 Cells-Derived Exosomes

Due to their advantages, which include rapid growth, simple culture and easy manipulation, HEK293 cells have also been widely used as cell models for the production of exosomes as drug carriers [185]. In view of the high transfection efficiency of this cell line, the biological modification of exosomes derived from HEK293 cells was mostly achieved through transfection. In recent years, many research teams obtained a large number of custom-produced designs for HEK293 cells using an optogenetically engineered exosome system (EXPLOR) [186] or EXOtic [187] devices developed for HEK293 cells. Therefore, HEK293 cells have been the main “new forces” of engineered exosomes.

## 6. Conclusions and Perspectives

Since the discovery of EVs, relevant studies have emerged in a continuous stream covering almost every aspect from identification and characterization to functional discovery and clinical applications. Here, we describe the biogenesis and release mechanism of exosomes and focus on how cargos (including proteins and miRNAs) enter exosomes through these processes and ways to utilize these potential cargo-sorting mechanisms to obtain bioengineered exosomes for clinical research and treatment. However, there are still many puzzling questions that are worth exploring for seeking answers [188].

First, there is an urgent need for a more scientific and accurate definition of various types of EVs. Due to the lack of specific characteristics of the heterogeneity of different exosomal subtypes, it has been difficult to separate and purify each exosome subgroup. Future work will be required to improve the techniques for the accurate purification and characterization of the diverse subpopulations of extracellular vesicles. For example, specific proteins that can more effectively distinguish EV subtypes need to be identified, and a solid foundation for in-depth study into the mechanism underlying the nonrandom distribution of functional cellular molecules in exosomes is needed.

Second, our current knowledge of EV biogenesis is incomplete. Regarding the subcellular origin of exosomes, the current mainstream view is that exosomes are formed through endosomal pathways, but a recent study showed that some exosomes may be formed by direct budding of the plasma membrane, and the efficiency of this direct budding may be much higher than that of the endosomal pathway [189].

Furthermore, exosomes can be secreted through ESCRT-dependent or ESCRT-independent pathways. These complex mechanisms are likely to be responses to the external environment and physiological state of the cell. This paradigm is similar to “compensation” mechanisms in cells; that is, the same exosome production mechanism cannot be simply activated in all cells under all conditions. When a cell in a certain state lacks the substances necessary for a synthetic pathway, another synthesis pathway is activated. Some people believe that these complex mechanisms are likely involved in different exosome subgroups. In other words, these complex mechanisms lead to the “diversity” of exosomes, and the synthesis of various exosomes follows unique pathways. A recent study identified two subgroups of exosomes (large exosome vesicles, Exo-L, 90–120 nm; small exosome vesicles, Exo-S, 60–80 nm) by using asymmetric flow field-flow fractionation (AF4) and found that these two subpopulations have unique N-glycosylation, protein, lipid, DNA and RNA profiles and biophysical properties, indicating that they have different biological functions [190].

In conclusion, EV science is moving forward with new discoveries and new technologies, such as single-vesicle analysis [191,192]. The novel function and properties of EVs will be revealed, and this knowledge will likely help resolve some questions. In addition, the development of EV biology is leading to a variety of strategies for the design and engineering of EVs as ideal drug carriers for future applications.

## Figures and Tables

**Figure 1 membranes-12-00498-f001:**
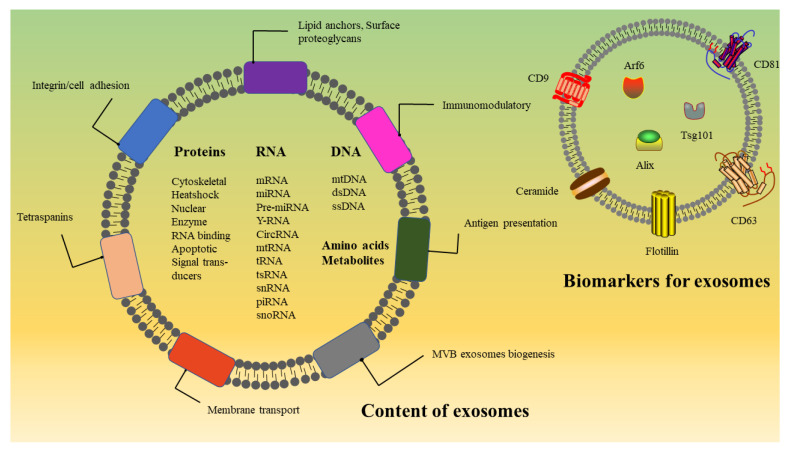
Structure and composition of exosomes. Exosomes carry various bioactive molecules and are important mediators of cell–cell communication. Exosome surface proteins mainly include tetraspanins, integrins and immunomodulatory proteins. In addition, exosomes contain different types of intracellular protein, RNA, DNA, amino acids, lipids and metabolites. Some proteins are involved in exosome biogenesis, including Rab GTPases, ESCRT proteins, and other proteins also used as markers for exosomes, such as CD9, CD81, CD63, flotillin, TSG101, ceramide and Alix.

**Figure 2 membranes-12-00498-f002:**
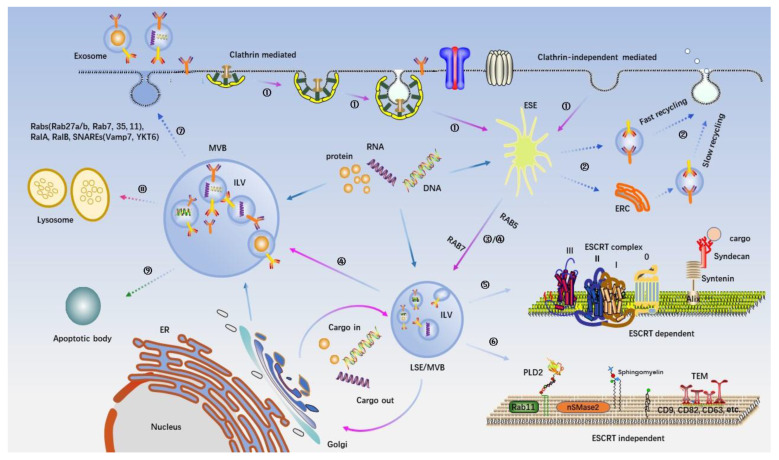
Intracellular machinery of exosome biogenesis and secretion. The origin and release of exosomes derived from eukaryotic cells are illustrated. Exosomes are formed as ILVs by the inward budding into early endosomes and MVBs. The first invagination of membrane can be achieved by clathrinid-mediated endocytosis (CME) and clathrinids-independent endocytosis (CIE) ①. After the early sorting endosome (ESE) is formed, part of it returns to the plasma membrane through “the fast recycling” and “the slow recycling” to complete the recovery of vesicles ②; the other part further develops into regular multivesicular bodies (MVB), which continue the transport of intracellular cargo ③. Subsequently, a second invagination of the membrane will occur on the MVB to form ILVs, which are the “precursors” of exosomes ④. The biogenesis mode of ILVs is very complicated and can be divided into two categories: an ESCRT complex-dependent pathway ⑤ and ESCRT complex-independent pathway ⑥. The former mainly involves multiple components of the ESCRT complex and the associated “Syndecan–Syntenin–ALIX pathway”. This approach is mainly responsible for recruiting ubiquitinated cargo and a few non-ubiquitin-modified cargoes to the restricted membrane of MVB and providing the physical structure and power of membrane invagination. The latter includes some lipids in membrane rafts and various tetraspanins, which together form a microdomain as a “pier” for cargo traffic. After MVBs containing multiple ILVs are formed, they need to face the final decision: fusion with the plasma membrane ⑦ or fusion with the lysosome ⑧ or apoptotic body ⑨. Some RAB proteins (RAB27a/b, RAB11, RAB7, RAB35) and some SNAREs (Vamp7, YKT6) have been identified to participate in the process of MVB and plasma membrane fusion. The specific mechanism involved is still not fully understood.

**Figure 3 membranes-12-00498-f003:**
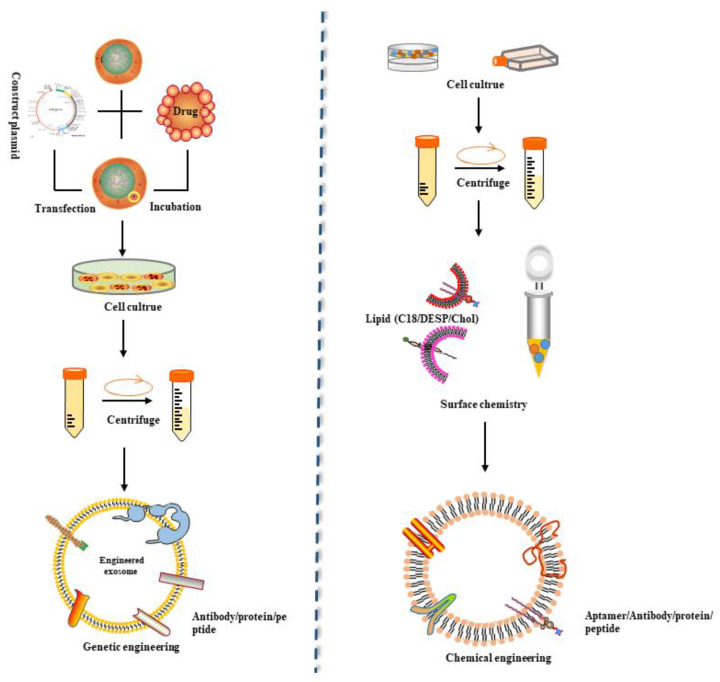
Schematic representation of the primary strategies for engineering exosomes displaying targeting ligands. Exosomes have the potential to serve as drug carriers engineered by different methods. Exosome modification mainly involves two methods: active approaches and passive approaches. The former includes methods of incorporating target substances in the process of exosomal biogenesis, such as transferring target peptide-expressing plasmids into cells to generate exosomes with target ligands. The latter includes methods of loading or combining exogenous substances after exosome secretion, such as electroporation or chemical conjugation.

**Table 2 membranes-12-00498-t002:** miRNA sorting mechanisms.

	Mechanism	References
nSMase2	miR-210 and miR-10b are incorporated into exosomes via a ceramide-dependent pathway	[67,68,69]
3′ end sequence of miRNAs	3′end urine glycation promotes the release of miRNA to exosomes	[70]
SP/NK-1R signaling	SP/NK-1R signaling increased the level of miR-21 in the exosome cargo.	[71]
hnRNPA2B1	SUMOylated hnRNPA2B1 binds miR-198 via the GGAG motif	[72]
hnRNPA1	Binds miR-196a and miRNA320 via potential UAGGUA/ AGAGGG to load into exosomes	[73]
SYNCRIP/hnRNP-Q	Packages miR-3470a and miR-194-2-3p into exosomes through its own NURR domain directly bind to GGCU motif	[74]
Argonaute 2	Packages let-7a, miR-100 and miR-320a into exosomes through KRAS–MEK–ERK signaling pathway	[75]
YBX-1	The interaction of YBX-1′s internal cold shock domain with miR-223	[76,77]
SRSF1	Binds miR-1246 via a 6 bp length motif (GG bases at positions 3 and 4)	[78]
MVP	forms an MVP protein-miR-193a complex	[79]
MEX3C	Sorts miR-451a by interacting with AP-2 (involved in exosome biogenesis)	[80]
La protein	Binds miR-122 via specific motifs, such as UGGA motif	[81]
FMR1	FMR1 is recruited to MVBs by cRILP and binds miR155 via AAUGC motif	[82]

## Data Availability

Not applicable.

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
