# Peer review of "Current Knowledge on Exosome Biogenesis, Cargo-Sorting Mechanism and Therapeutic Implications"

_membranes, 2022, doi:10.3390/membranes12050498_

Round 1
Reviewer 1 Report
The authors have written a methodical and comprehensive review of proteins involved in the formation of and for sorting cargo into exosomes. In addition, they have reviewed approaches to load, potentially therapeutically beneficial, cargos into exosomes for use as biotherapeutics. I believe this review will be a helpful addition to the field.
Reviewer 2 Report
The authors in this review described in the first part the mechanisms involved in exosomes formation and release, while in the second part they focused their attention on bioengineering of exosomes and their therapeutic use. The review is well written and it easily to follow although it contains many different information.
However, this manuscript can be accepted after major and minor revision.
Major:
- The title does not reflect the content along the manuscript; indeed, it summarize only the first part of this manuscript (chapter 1 to 3). The authors should change it accordingly. Moreover, in the title the authors refer to extracellular vehicles but along the manuscript they refer to exosomes. Since these two worlds are not synonymous, I suggest to authors to uniform the nomenclature.
- The figure 1 is very self-explanatory. I suggest including an extra figure that describe the “release of exosomes” in order to summarize the multitude of studies described in this chapter.
- The authors in the introduction part they did not mention the term of extracellular vesicles (EVs), but the chapter 2 is fully dedicated to the description of them. They should add a sentence at the end of the introduction paragraph that briefly explain the term EVs. Moreover, in the chapter 3 the title includes the acronym “EVs” but within this specific they did not mention them but only the exosomes. Please correct accordingly.
Minor:
- Line 104, extracellular vesicles appear for the first time, it needs the acronym (EVs).
- In line 123, they stated that the EVs isolated by ultracentrifugation can be contaminated by coprecipitation of EVs component. Since it is not an affirmation due to the findings within the manuscript the authors should include the reference.
- In the table 1, the authors should explain the acronym ESCRT. I suggest including at the end of the table to add the abbreviations used.
- In line 230 the world “morpholo-gical” is truncated. Please correct it
- In line 458 it misses the capital letter “I” at the beginning of the sentence.
- Line 633 the worlds “protein” and “was” are attached. Please correct the typo
Round 2
Reviewer 2 Report
The authors addressed all my comments.
I have no other requests